# Natural Variation Uncovers Candidate Genes for Barley Spikelet Number and Grain Yield under Drought Stress

**DOI:** 10.3390/genes11050533

**Published:** 2020-05-11

**Authors:** Samar G. Thabet, Yasser S. Moursi, Mohamed A. Karam, Andreas Börner, Ahmad M. Alqudah

**Affiliations:** 1Department of Botany, Faculty of Science, University of Fayoum, Fayoum 63514, Egypt; sgs03@fayoum.edu.eg (S.G.T.); ysm01@fayoum.edu.eg (Y.S.M.); mak04@fayoum.edu.eg (M.A.K.); 2Research Group Resources Genetics and Reproduction, Department Genebank, Leibniz Institute of Plant Genetics and Crop Plant Research, 06466 Seeland OT Gatersleben, Germany; boerner@ipk-gatersleben.de

**Keywords:** GWAS, drought, barley, spikelet development, candidate gene

## Abstract

Drought stress can occur at any growth stage and can affect crop productivity, which can result in large yield losses all over the world. In this respect, understanding the genetic architecture of agronomic traits under drought stress is essential for increasing crop yield potential and harvest. Barley is considered the most abiotic stress-tolerant cereal, particularly with respect to drought. In the present study, worldwide spring barley accessions were exposed to drought stress beginning from the early reproductive stage with 35% field capacity under field conditions. Drought stress had significantly reduced the agronomic and yield-related traits such as spike length, awn length, spikelet per spike, grains per spike and thousand kernel weight. To unravel the genetic factors underlying drought tolerance at the early reproductive stage, genome-wide association scan (GWAS) was performed using 121 spring barley accessions and a 9K single nucleotide polymorphisms (SNPs) chip. A total number of 101 significant SNPs, distributed over all seven barley chromosomes, were found to be highly associated with the studied traits, of which five genomic regions were associated with candidate genes at chromosomes 2 and 3. On chromosome 2H, the region between 6469300693-647258342 bp includes two candidate drought-specific genes (*HORVU2Hr1G091030* and *HORVU2Hr1G091170*), which are highly associated with spikelet and final grain number per spike under drought stress conditions. Interestingly, the gene expression profile shows that the candidate genes were highly expressed in spikelet, grain, spike and leaf organs, demonstrating their pivotal role in drought tolerance. To the best of our knowledge, we reported the first detailed study that used GWAS with bioinformatic analyses to define the causative alleles and putative candidate genes underlying grain yield-related traits under field drought conditions in diverse barley germplasm. The identified alleles and candidate genes represent valuable resources for future functional characterization towards the enhancement of barley cultivars for drought tolerance.

## 1. Introduction

Barley (*Hordeum vulgare* L.) ranks as one of the most important cereal crops worldwide. Globally, barley is the fourth most important cereal crop in terms of production after maize (*Zea mays* L.), rice (*Oryza sativa* L.), and wheat (*Triticum spp.*) (Faostat 2017, http://www.fao.org/faostat/en/#home). One limitation in achieving the production target is abiotic stress which limits the quality and nutritional value of the grain in cereal crops worldwide [1]. Among all abiotic stresses, drought is the most important environmental stress which limits crop production and yield [2,3,4], and is becoming more common particularly in the arid and semiarid regions [2].

Crops can be exposed to drought during their entire life cycle from vegetative to reproductive stages [5]. Drought stress affects crop growth and yield during all developmental stages [6]. Water shortage at early growth stages can cause severe problems for seedlings, restricting the emergence, growth and development of seedlings, and thus affecting grain yield [7]. Furthermore, the developing plants will have poor tillering capacity, leading to fewer tillers per unit area and thus lower yield potential. Moreover, drought in the period of stem elongation causes a decrease in the number of grains per unit area because it has a negative impact on floret formation and fertility [8].

At post-anthesis, water insufficiency reduces the grain filling rate and duration leading to shriveled grains [2]. Moreover, the effect of drought on yield is highly complex and involves processes as diverse as reproductive organs, gametogenesis, fertilization, embryogenesis and seed development [6,9]. Reproductive and seed development phases are especially sensitive to drought stress [2,10]. In barley, a reduction in the number of grains per spike, grain filling duration and dry matter accumulation have already been reported to decrease grain yield [2,10]. Several studies have reported that early growth stage parameters (e.g., tiller number, biomass formation, etc.) are highly correlated with yield potential and grain quality at harvest under both normal and drought stress conditions in various cereal crops, including barley [11,12]. Accordingly, understanding the genetic basis for drought tolerance in crop plants by identifying the genetic loci and the candidate genes associated with these traits is useful for developing new varieties with more drought-tolerant characters.

The Genome-Wide Association Scan (GWAS) approach is widely employed to reveal associations between genomic loci and advantageous traits in a given population based on linkage disequilibrium [13]. These loci then become targets for improving new genotypes by the breeder. The GWAS is very effective in identifying major candidate genes regulating mono- or oligogenic agricultural traits. Recently, GWAS has been successfully used to identify genes for yield-related traits [14,15]. Barley has a high-level population structure such as two-rowed and six-rowed cultivars, spring and winter barley [16]. This may lead to spurious marker–trait associations in GWAS [17]. Therefore, it is important to use strong statistical methods and strategies to control the population structure [17]. A mixed-linear model (MLM) approach has been developed to control spurious associations through account multiple levels of relatedness leading to better performance [18].

GWAS has successfully yielded genomic locations for quantitative trait loci (QTL) in crop cereals [13]. Identification of genomic locations for QTL using linked segregating markers is considered to be highly useful for marker-assisted breeding. Nevertheless, the ultimate goal of GWAS in crop species is to detect new QTL through genetic dissection of complex traits.

The present study aimed at identifying the genetic basis of drought tolerance at different developmental and growth phases in 121 spring barley accessions under field conditions using GWAS. In total, 101 SNPs showed association with different traits that were distributed across the seven chromosomes of barley. The identified QTL colocalized with several genes that are exclusively distributed on chromosomes 2H and 3H. The annotation and expression of these genes demonstrated their roles in drought tolerance.

## 2. Materials and Methods

### 2.1. Experimental Setup and Phenotyping

In total, 121 diverse spring barley accessions from different geographical origins were grown under the field conditions in the 2017/2018 growing season at the Experimental Station of University of Fayoum. The collection included 83 cultivars, 29 landraces, and 9 breeding lines. They originated from Europe (EU, 62), West Asia and North Africa (WANA, 24), East Asia (EA, 22), and the Americas (AM, 13). The row types were two-rowed (72) and six-rowed (49). The population structure and genetic diversity using the genotypic information of the accession in the collection are shown in Appendix A that demonstrated there is no clear structure on our collection.

Five seeds from each accession were sown in plastic pots (40 cm × 26 cm × 26 cm) filled with field-soil on the 1st of December 2017 under field conditions. The soil texture was classified as clay-loam with pH = 8.0, total N% = 9.4 and available P = 58.0 ppm. Manual irrigation was performed as required, and 5 g (17:11:10/N:P:K) fertilizer was added to each pot. In field-grown plants, each accession was replicated in three pots of each treatment (control and drought). At the beginning of early reproductive (spikelet development phase) phase (~25 days after sowing), the plants were thinned into three plants per pot, standing with a border to eliminate positional and environmental effects on growth and development. Weeds were controlled manually.

Plants were irrigated until the onset of early reproductive, then the plants were then exposed to two watering treatments: (1) well-watered treatment (soil maintained at ~75% of field capacity (FC)); and (2) severe drought stress (at 35% FC). To maintain the targeted (~75% FC) and drought (~35% FC), eight randomly selected accessions were used as a reference. Before irrigation, eight reference pots were weighed and watered to adjust the corresponding field capacity, and the rest of the experiment was watered accordingly. Irrigation and drought treatment continued until maturity after that, irrigation withholds until harvest. Nine morphological, developmental and grain yield-related traits were measured from at least three biological replicates for each accession under each treatment. More information about the phenotypic trait measurements are explained in Table 1. Respective drought tolerance indices were calculated from the recorded data as described in (Table 1).

The average of temperature and humidity during the growing season of this experiment (2017–2018) was 16.62 °C and 55.16%, respectively. The maximum temperature was recorded in November and April (24.33 and 27.66 °C, respectively) with the minimum temperature from January to February (8.29 and 9.12 °C, respectively). The minimum humidity was 30.37% and 30.41% in November and February, respectively, whereas the maximum humidity was recorded in April (78.33%) (Appendix A).

### 2.2. Data Analysis

Analysis of variance [19] was conducted to compare the controlled and drought stress conditions at *p* < 0.05 for all measured traits using GENSTAT 18 [20]. Data were analyzed as a randomized complete block design (RCBD) with three replications. The treatments were considered as main plots and the accessions were considered as sub-plots. Broad-sense heritability (*H*^2^) for the measured traits under each condition separately was calculated using GENSTAT 18. The phenotypic data were subjected to Residual Maximum Likelihood (REML) to analyze it in mixed linear model (MLM). Mean estimation of each measured trait in each accession under each treatment was calculated as Best Linear Unbiased Estimates (BLUEs) using GENSTAT 18. Correlation matrix analysis among the traits in each treatment was separately calculated by GENSTAT 18. Comparison between treatments at *p* < 0.05 for each trait including boxplots were calculated using R-studio [21].

### 2.3. Genome-Wide Marker–Trait Associations

The accessions were genotyped by a 9K Illumina^TM^ SNPs chip. In the analysis, we only used the markers which passed the quality control as minor allele frequency (MAF) ≥ 0.05 with their physical positions. A mixed linear model (MLM) was performed to determine marker–trait associations between the estimated phenotypic traits (BLUEs) and genotypic data. Different statistical models, e.g., general-linear model (GLM), mixed linear model (MLM), and compressed MLM (CMLM)) were tested in GWAS using GAPIT R package [22]. Finally, we used MLM as a powerful model considering the population structure, including kinship and PCA, to control the population structure influence. False discovery rate (FDR) at 0.001 was calculated for each trait under each treatment separately and association signals passed the threshold of FDR at 0.001 (−log_10_
*p*-values ≥ FDR) were used in further analyses. To be sure that our associations were true, we followed the GWAS and post-GWAS protocol published recently [13].

### 2.4. SNP-Gene-Based Association and Haplotype Analysis

At each chromosome, linkage disequilibrium (LD, r^2^) among the significant SNPs within highly associated genomic region was calculated and presented as heatmap plot. This allowed us to define the most important physical position that had been used for candidate gene identification. The physical positions of SNPs exceeding FDR within the linkage disequilibrium interval were used in annotation for high-confidence (HC) candidate gene with other respective information using the barley genome explorer web-based with recent barley genome dataset (BARLEX; http://apex.ipk-gatersleben.de).

SNPs within the candidate gene physical position were used for further validation of SNP-Gene-based haplotype analyses and expression analyses. *T*-test at *p* < 0.05 was used to calculate the significant differences between alleles on the associated trait(s) [13]. RNA-seq datasets were derived from 16 different tissues of barley cv. ‘Morex’ cultivar, each with three biological replicates. In total 48 samples were used for generation of RNA-seq data.

From seven vegetative, six inflorescence, two developing grain and one germinating grain tissues, more details about RNA-seq experiments was published by Mascher et al. [23]. We used BARLEX; an expression database for barley that presented as FPPM (fragments per kilobase per million mapped reads).

## 3. Results

### 3.1. Phenotypic Characteristics and Natural Variation

In total, nine traits were recorded under control and drought treatments. Additionally, respective drought tolerance indices were calculated and used as a derived trait for GWAS. A wide range of phenotypic variation with normal distribution was detected for all traits (Figure 1 and Appendix A). The means of most of the studied traits showed a significant reduction under drought treatment compared to control conditions (Table 2 and Appendix A). There were no significant differences between the treatments in phase transition i.e., AT, SH and AE developmental stages (Table 2 and Appendix A). Drought treatment influenced significantly other developmental and yield traits such as plant height and spikelet number per spike (Table 2 and Appendix A). Notably, the genotypes showed wide range of variation in all drought tolerance indices (Table 2, Figure 2 and Appendix A).

Furthermore, heritability values were relatively high under drought, ranging from 0.64 for TGW to 0.84 for AT, whereas they ranged from 0.72 to 0.80 for AT and SH, respectively under control conditions. Additionally, the heritability values varied for tolerance indices from 0.68 for AE_DTI and NGS_DTI to 0.78 for AT_DTI (Table 2).

### 3.2. Correlations Analysis

Significant positive correlations were observed among various traits within both treatments. For example, AE showed a significantly high positive correlation with AT and SH (*r* = 0.98 *** and *r* = 1.00 ***, respectively) under control conditions and (*r* = 0.93 *** and *r* = 1.00 ***, respectively) under drought treatment (Figure 3A,B and Appendix A). On the contrary, some traits showed high significant negative correlations under both conditions. Interestingly, TGW showed negative correlation with NGS under control and drought conditions (*r* = −0.21 and −0.44 ***, respectively; Figure 3A,B).

For DTI, AT_DTI showed high significant positive correlation with SH_DTI and AE_DTI (*r* = 0.92 *** and *r* = 0.91 ***, respectively). Additionally, SH_DTI exhibited high positive correlation with AE_DTI (*r* = 0.99 ***). The SL_DTI with AL_DTI showed a significant negative correlation (*r* = −0.27 ***; Appendix A).

### 3.3. Natural Genetic Variation and Candidate Genes Potentially Underlying Drought Tolerance

GWAS analysis of 121 selected accessions was performed to find out the natural genetic variation of the studied traits. We detected a total number of 101 significant marker–trait associations (with −log_10_
*p*-value ≥ 3) distributed over the seven barley chromosomes (Figure 4 and Appendix A). There was plenty of natural genetic variation of all studied traits in this collection. Through GWAS analysis, we found twelve interesting genomic regions for genetic variation of all studied traits distributed on only two chromosomes (2H and 3H).

Highly significant LD was found among significant SNPs within these genomic regions (Figure 4b), indicating that these significant SNPs are potentially harboring important candidate genes in addition to their useful for marker-assisted selection. On chromosome 2H, two adaptive genes were identified, whereas on chromosome 3H, ten genes that represent a combination of both adaptive and constitutive genes were identified. Adaptive genes might be control-specific, i.e., genes that regulate trait variation under control only, or drought-specific (genes that regulate trait variation under drought only). Constitutive genes regulate trait variation under both control and drought conditions (Table 3).

In total, eight SNPs were associated (−log_10_
*p*-value ≥ 3) with AT parameters. Out of these, five SNPs were adaptive: three were control-specific and two were drought-specific; the remaining three SNPs were constitutive. The constitutive SNPs were identified on chromosomes 1H, 3H and 4H. The most significant one (with −log_10_
*p*-value = 8.7) was observed on chromosome 3H at 126.69 cM. Only one constitutive gene was identified *HORVU3Hr1G098200* (Chr. 3; 126.69 cM) (Table 3).

Sixteen SNPs were associated (−log_10_
*p*-value ≥ 3) with SH parameters. Out of these, seven SNPs were adaptive: four SNPs were control-specific and three drought-specific; the remaining nine SNPs were constitutive. The constitutive SNPs were identified on chromosomes 1H, 2H, 3H, 4H, 5H and 6H where the most significant one (with −log_10_
*p*-value = 4.7) was observed on chromosome 6H at 53.54 cM. Only seven SNPs showed association with candidate genes. Four genes were control-specific; *HORVU3Hr1G088300*, *HORVU3Hr1G089160*, *HORVU3Hr1G089080* and *HORVU3Hr1G098200*. Three constitutive genes were identified: *HORVU3Hr1G098200*, *HORVU3Hr1G116790* and *HORVU3Hr1G115810* (Table 3).

In total, 18 SNPs showed association (with −log_10_
*p*-value ≥ 3) with AE parameters. Ten SNPs were adaptive: six were control-specific and four were drought-specific; the remaining SNPs were constitutive. These constitutive SNPs were mapped on chromosomes 2H, 3H, 4H, 5H and 6H. The most significant one (with −log_10_
*p*-value = 4.6) was detected on chromosomes 6H at 53.54 cM. Out of these, twelve SNPs showed association with candidate genes. Eight genes were control-specific: *HORVU3Hr1G018650*, *HORVU3Hr1G020430*, *HORVU3Hr1G020660*, *HORVU3Hr1G019590*, *HORVU3Hr1G088300*, *HORVU3Hr1G089160*, *HORVU3Hr1G089080* and *HORVU3Hr1G098200* and four genes are constitutive: *HORVU3Hr1G116790*, *HORVU3Hr1G098200* and *HORVU3Hr1G115810* (Table 3).

For the trait PH, six SNPs were associated. Out of these, five SNPs were adaptive: four SNPs were control-specific and one SNP was drought-specific. Only one constitutive SNP was found on chromosome 7H at 131.59 cM. There were no SNPs associated with candidate genes for PH (Table 3).

GWAS analysis showed that six SNPs were associated with SL parameters. Only one SNP was associated with a candidate gene that was control-specific, *HORVU3Hr1G098200* (Table 3).

In total, eight SNPs showed association (*p*-value ≤ 0.001) with AL. Out of these, six SNPs were adaptive: two were control-specific and four were drought-specific; the remaining two SNPs were constitutive and mapped on chromosomes 2H and 3H. Six SNPs were associated with candidate genes. Only one gene was control-specific, *HORVU3Hr1G098200*; and one was constitutive, *HORVU3Hr1G098200* (Table 3).

For NSS, 15 SNPs showed significant association with −log_10_
*p*-value ≥ 3. Six of them were associated with adaptive candidate genes. Three SNPs were control-specific: *HORVU3Hr1G018650*, *HORVU3Hr1G020430* and *HORVU3Hr1G020660* and another three were drought-specific: *HORVU2Hr1G091030*, *HORVU2Hr1G091170* and *HORVU3Hr1G019590*. No constitutive genes were identified (Table 3).

In total, 15 SNPs were associated with NGS parameters. Seven SNPs showed association with candidate genes. Five genes were control-specific: *HORVU2Hr1G091030*, *HORVU3Hr1G018650*, *HORVU3Hr1G020430*, *HORVU3Hr1G020660* and *HORVU3Hr1G019590* and two drought-specific genes: *HORVU2Hr1G091030* and *HORVU2Hr1G091170* (Table 3).

For TGW parameters, seven SNPs showed significant association with −log_10_
*p*-value ≥ 3. Only two SNPs showed association with candidate genes. One was control-specific, *HORVU3Hr1G098200*, and the other drought-specific one was *HORVU3Hr1G098200* (Table 3).

Overall, all the identified genes revealed a pleiotropic effect, i.e., each gene controlled more than one trait. The gene *HORVU3Hr1G098200*, for instance, regulates the variation of ten traits (Table 3). On the contrary, they differ in their mode of action, and some of them are adaptive genes such as *HORVU2Hr1G091170* and *HORVU3Hr1G018650*. The first gene modulates the variation of NGS and NSS under drought; whereas the second controls them under control. Other genes are constitutive, such as *HORVU3Hr1G116790* and *HORVU3Hr1G115810* (Table 3). Surprisingly, the gene *HORVU3Hr1G098200* showed a constitutive/adaptive mode of action. It controls the variation of (TGW) constitutively, while regulating the variation of (AE, AL, SH and SH) in an adaptive manner, i.e., under control only.

### 3.4. SNP-Gene-Based Analysis

Twelve SNPs at chromosomes 2H and 3H were physically co-located inside the candidate genes (Table 3). Two SNPs at 2H (SCRI_RS_166540 and SCRI_RS_157347) were detected within the genes *HORVU2Hr1G091030* and *HORVU2Hr1G091170*, respectively. Meanwhile, ten SNPs at 3H were co-located within the physical position of candidate genes. For example, SNP numbers BOPA1_2391-566 and SCRI_RS_177313 were located within *HORVU3Hr1G019590* and *HORVU3Hr1G098200* genes, respectively. Interestingly, these SNPs were mostly associated with NGS, NSS, and TKW under drought stress. The rest of the genes at 3H were associated with traits under drought stress or with DTI for SH and AE.

The allelic analysis of the SNPs that are associated with traits under drought showed that alleles A, G and A from markers SCRI_RS_166540, SCRI_RS_157347 and BOPA1_2391-566, respectively, have a highly significant impact on NSS (Figure 5a and Figure 6). The genes *HORVU2Hr1G091030* and *HORVU2Hr1G091170* at 2H controlled NGS under drought via markers SCRI_RS_166540 and SCRI_RS_157347, where alleles G and A, respectively, increased NGS significantly (Figure 5b and Figure 6). Only one marker (SCRI_RS_177313 from *HORVU3Hr1G098200* gene) showed a significant effect on TKW by allele A that increased the value under drought (Figure 5b).

### 3.5. Expression Analysis of Candidate Genes

The expression analysis of candidate genes in different organs showed a wide range of expression for the genes (Figure 6). Notably, the associated genes with spikelet and grain number per spike under drought stress (Figure 5) showed a high expression for most of the organs (Figure 6). Gene *HORVU2Hr1G091170* at 2H in particular that had high impact on spikelet and grain numbers under drought showed highest expression in the respective grain organs, e.g., lodicule and rachis in addition to spike at 1–1.5 cm length (Figure 6). The second highest expressed gene from the highly associated ones was *HORVU2Hr1G091030*. This gene was highly expressed in developing grain and lodicule (Figure 6). The expression of these genes demonstrated their biological roles in the spikelet and grain development under drought conditions. Other genes, e.g., *HORVU3Hr1G020660* and *HORVU3Hr1G018650*, showed high expression particularly in senescing leaves, suggesting their roles in leaf development (Figure 6).

## 4. Discussion

Studying drought stress tolerance under field conditions in cereals such as barley is very limited, as it requires complex and laborious experiments for population characterization, in addition to being influenced by environmental factors [3,4]. Nevertheless, the present study focused on investigating the natural variation in diverse spring barley collections and on identifying candidate genes associated with the traits of interest under field conditions.

In the present study, there was a considerable reduction in most traits under drought stress compared to control conditions. These results indicated that drought stress reduced grain yield by decreasing NSS, NGS, and TGW. Our results are supported by the findings of [2,24], who examined the response of spring barley to pre- and post-anthesis drought and reported a yield reduction due to pre-anthesis water deficit on several fertile spikes and grains per plant. Our results are in agreement with those of Samarah, Alqudah, Amayreh and McAndrews [2], who found that drought stress reduced grain yield by reducing the number of tillers, spikes and grains per plant and individual grain weight in barley (*Hordeum vulgare L.*) as a result of early maturity and shortened grain filling duration at 25% field capacity compared to control. In conclusion, drought stress negatively influenced barley yield through impairing grain development, size and grain filling duration.

Agronomic traits such as grain yield and its components (NSS, NGS, and TGW) are the major selection criteria for drought tolerance in barley breeding [25]. Therefore, understanding the interplay among these parameters is of high importance. The correlation between TGW and NGS was always negative and significant under control and drought, respectively. The positive and significant correlation between TGW and SL indicates that yield mainly depends on spike length. Zhou et al. [26] suggested that the grain yield traits interacting with each other, increase in one of them (e.g., TGW) can be correlated with a reduction in another (e.g., NGS), which is in agreement with our results. In support of our findings, in a European spring barley collection [27], it was found that NGS showed negative correlations with all other yield parameters except grain length, concluding that the yield was mainly dependent on grain size and SL rather than NGS. Several reports showed that drought-tolerant genotypes implement high productivity under both stressed and unstressed conditions [2,28]. Therefore, the comparative analysis of the yield components under drought and well-watered conditions can be used for predicting stress tolerance of genotypes, and then in selection of more tolerant barley lines for future breeding purposes [24,29].

### The Role of Putative Candidate Genes in Drought Tolerance

Promising genomic regions are located at position 75.56 cM on chromosome 2H, harboring two important candidate genes. The first one is *HORVU2Hr1G091030*, which encodes RNA polymerase II C-terminal domain phosphatase-like 1 (*CPL1*), for NGS_C, NGS_D, and NSS_D. *CPL1* is a negative regulator of stress-responsive gene transcription, ABA, and stress responses [30]. In Arabidopsis, *CPL1* regulates gene expression under various osmotic stresses through ABA signaling [30]. Loss of *CPL1* function in the mutants enhances tolerance to oxidative stress including drought and salt stresses [31]. In rice, *OsCPL1* is expressed in the young spikelets [32]. Most likely, this gene is expressed during the development of barley spikelets under drought. Additionally, it controls the NGS under drought and control conditions in a constitutive manner. The second candidate, *HORVU2Hr1G091170*, encodes expansin B3 for NSS_D and NGS_D. Expansins have been implicated in the responses of various plant species to water stress. In maize, increased expansin activity was found to be involved in maintaining the growth of primary roots at a low water potential [33]. The expression of a root β-expansin gene, *GmEXPB2*, is remarkably participated in root system architecture responses to several abiotic stresses, such as Fe, P, and water deficiency [34]. *RhEXPA4* is a rose expansin gene that is up-regulated in rose petals after dehydration [35], and it confers salt and drought tolerance to transgenic Arabidopsis [36]. In potato, most of expansin-like B genes have a potential role in multistress tolerance and upregulated under stresses including drought [37]. These two genes—*HORVU2Hr1G091030* and *HORVU2Hr1G091170*—showed different expression profiles (Figure 6). The noisy expression profile of *HORVU2Hr1G091170*, suggesting a stress-responsive gene, exclusively regulates the variation of NGS_D and NSS_D under drought (Table 3). While the slightly constant expression profile of *HORVU2Hr1G091030* indicates a constitutive gene that regulates the variation of NGS under control and drought. These findings are in accordance with the results of several authors who found that the stress-responsive genes exhibiting noisy expression profiles, while the constitutive ones showing constant expression patterns reviewed in Lopez-Maury et al. [38].

Notably, the allelic diversity analysis shows that allele A and G from *HORVU2Hr1G091030* and *HORVU2Hr1G091170* genes, respectively, have a significant positive impact on NSS and then NGS under drought stress. The expression of these genes during spike, spikelet and grain developmental stages demonstrates their influence on agronomic traits under drought stress conditions. In terms of molecular breeding, the above-mentioned alleles were the highest alleles explained the natural variation under drought stress suggesting their usefulness in breeding programs. Taken together, this provides evidence that these genes are drought-specific and involved in the drought stress-tolerance pathway. These findings indicate that both genes are of high importance for enhancing barley grain yield under drought stress.

Interestingly, ten candidate genes were detected on chromosome 3H. Out of these, three genes at positions 44.26 and 45.55 cM were identified as candidates for various traits such as AE, NGS, and NSS under control conditions. The first one is *HORVU3Hr1G018650* encoding pyruvate decarboxylase-2 (*PDC2*), which belongs to pyruvate decarboxylase (*PDC*) gene family. Pyruvate has been involved in the ethanolic fermentation pathway that is associated with flooding tolerance when plant cells switch from respiration to anaerobic fermentation [39]. Additionally, fermentation has important functions in the presence of oxygen, mainly in germinating pollen and during abiotic stress. This indicates the interdependency between pyruvate decarboxylase (PDC) and AE, NGS, and NSS under control conditions. Furthermore, PDC, which catalyzes the first step in this pathway, is thought to be a main regulatory enzyme [40]. In Arabidopsis, the expression of PDC genes during abiotic stresses has been reported [41]. In maize and Arabidopsis, strong induction of fermentation genes takes place in anaerobic conditions [42]. Thus, it is conceivable that ethanolic fermentation is part of a general response to environmental stress, e.g., drought stress.

For traits such as AE_C, NGS_C, and NSS_D at position 46.25 cM, *HORVU3Hr1G019590* encodes myb domain protein 37. MYB (myeloblastosis) has a regulatory role in ABA signaling by activating some stress-inducible genes [43]. In Arabidopsis, MYB, namely *AtMYB60*, *AtMYB44*, and *AtMYB15*, have been implicated in the regulation of stomatal closure and ABA-mediated response to drought and salt stresses [44]. Agarwal et al. [45] detected that *AtMYB15* was expressed in both vegetative and reproductive organs and up-regulated by cold and salt stresses. The differential expression of numerous MYB TFs in the *Triticeae* was shown to be involved in the response of abiotic stress conditions such as drought and salt stresses [46,47]. These findings suggest that the genes in question are constitutive genes and may have a role in drought tolerance in barley during heading and maturation.

For SH_C and AE_C, *HORVU3Hr1G089160* encodes AP2-like ethylene-responsive transcription factor at position 104.32 cM. The AP2/ERF superfamily regulates diverse developmental responses such as flower pedicel abscission [48], leaf senescence [49], cell proliferation and shoot branching [50]. Houston et al. [51] observed that mutants of *HvAP2* internode lead to the reduction of elongation in both the culm and the spike in barley, suggesting that the *HvAP2* alleles increase grain yield by controlling spike density. In the current study, the allelic variation was only observed under control condition indicating that this gene might not be involved directly in drought tolerance.

Additionally, *HORVU3Hr1G116790* encodes Aquaporin-like superfamily protein which is a candidate for SH_DTI and AE_DTI. Aquaporins (AQPs) are a class of water channel proteins that belong to the major intrinsic protein (MIP) superfamily of membrane proteins [52]. These proteins regulate the movement of water and other small molecules across plant vacuolar and plasma membranes [53]. Aquaporins have also been suggested to have an essential role in plant tolerance of biotic and abiotic stresses [54], and extension growth [52]. Furthermore, it was reported in [55] that various aquaporin homologs are involved in plant stress responses against a variety of environmental stresses that disturb plant cell osmotic balance and nutrient homeostasis.

The most effective gene *HORVU3Hr1G098200* was mapped at 126.69 cM. This gene orchestrates the variation of ten traits both in constitutive and adaptive manner (Table 3). This explains the significant correlations between these traits, either under control or drought (Table 3 and Figure 3A,B). Shi et al. [56] reported that Chromosome 3B harbors genes that may be significant in controlling agronomical important traits, such as yield and resistance to biotic and abiotic stress in wheat. The QTL for these traits maps quite close to semidwarf1 (sdw1) on chromosome 3H at 126.69 cM. The *semi dwarf 1 (sdw1)* gene has previously been found to control the most desired agronomic traits barley reviewed by Hedden [57]. The pleiotropic effect of *sdw1* gene was evidenced in wheat [58,59].

Additionally, the gene *HORVU3Hr1G098200* showed an interplay between the constitutive and adaptive control pattern. For example, it constitutively controls the variation of thousand grain weight (TGW) under both control and drought stress. At the same time, it controls the spike heading only under control in an adaptive manner (Table 3). This pattern indicates that *HORVU3Hr1G098200* is partially constitutive and partially stress-responsive gene. Therefore, we considered it an adaptive gene when it controlled a trait(s) under control or drought. On the other hand, we considered it constitutive when it controlled trait(s) under both conditions. This gene, nevertheless, exhibited constant expression profile in different plant tissues, spanning the reproductive period from anther extrusion until the seed set, suggesting a key role in controlling different traits and more likely to be constitutive (Figure 6). This finding is consistent with the results of [60], who found that cells express some genes constantly to maintain the concentration of some proteins tuned with the cell physiological needs. The constant expression pattern characterizes the constitutive genes rather than the stress responsive (adaptive) ones (reviewed in Lopez-Maury, Marguerat and Bahler [38]. According to Blum [61], drought stress when expressed as a final yield is affected by constitutive and adaptive plant traits (genes). These constitutive QTL/genes represent an instrumental tool for selection as they show stability across different environments compared to the adaptive ones. Moreover, the selection for drought tolerance based on these QTL/genes does not require drought stress.

The last candidate gene is *HORVU3Hr1G115810*, which encodes Kinetochore protein spc25 and affects AE_DTI and SH_DTI. Kinetochore proteins may have a pivotal role for centromere and kinetochore functioning [62,63,64], and chromosome segregation mediating [65]. Specifically, kinetochore protein MIS12 is required for the co-orientation of sister kinetochores during meiosis I in maize [66]. NDC80 kinetochore protein serves as a contact point for chromosome–spindle interaction [67]. Interestingly, QTL at 3H 126 and 154 cM have previously been reported to be associated with grain number and yield in barley under drought stress conditions [4,68,69].

On chromosome 3H, three genes (*HORVU3Hr1G020660*, *HORVU3Hr1G088300*, and *HORVU3Hr1G098200*) are counterparts of the ortholog in Arabidopsis, *AT2G36270*. The corresponding genes are expressed during the reproductive stage in the different flower and seed organs (Figure 6), indicating their significance in flowering and seed set. Our results are in accordance with the findings of Klepikova et al. [70], who compared the spatiotemporal expression and stability of a lot of genes in 79 organs and developmental stages. Additionally, they found that *AT2G36270* had been significantly expressed in senescent organs (leaves and silique); this is similar to expression profile of the gene *HORVU3Hr1G020660* that highly expressed in senescent leave (Figure 6). Taking these findings together, the similarity of spatiotemporal expression patterns, as well as functions of these genes in both barley and Arabidopsis, suggests that they might be are homologous for AT2G36270.

The significant role of *AT2G36270* (*ABI5*, *AtABI5* and *BZIP39*) during different growth stages, as well as in drought tolerance, has been evidenced in several studies. Finkelstein et al. [71] reported that it encodes a member of the basic leucine zipper transcription factor family, involved in ABA-regulated gene expression during germination, seed development and subsequent reproductive stage. In particular, *ABI5* regulates a set of the late embryogenesis-abundant genes during both seed and ABA-inducible vegetative gene expression in wild-type and abi5-1 plants [71]. Mittal et al. [72] reported that overexpression of *AtABI5* in transgenic cotton (*Gossypium hirsutum*) showed resistance to the imposed drought stress through ROS scavenging and osmotic adjustment, enhancing photosynthesis, as well as traits of drought avoidance (bigger root and leaf systems) and tolerance (longer internode length and higher stem weight) leading to better establishment under water shortage. In rice (*Oryza sativa*), overexpression of *OsbZIP46CA1* significantly increased tolerance to drought and osmotic stresses at flowering stage, and suggested that *OsbZIP46* is a positive regulator of ABA signaling and drought stress tolerance by modulating many stress-related genes [73].

In summary, the present study showed the value of using field experiments to investigate natural phenotypic and genetic variation in a worldwide panel of barley accessions underlying agronomic traits such as grain yield and its components which can be exploited for crop improvement. Drought stress negatively influences most of the studied traits. In addition, we observed significant positive correlations among various traits within both control and drought treatments. Candidate genes associated with drought response were detected on two chromosomes, notably 2H and 3H. Interestingly, most the candidate genes are described to be involved in responses to abiotic stresses such as drought and salt. Interestingly, the genomic regions at 75 cM on 2H and 126.69 cM on 3H harbor three candidate gene *HORVU2Hr1G091030 HORVU2Hr1G091170* and *HORVU3Hr1G098200*, which are highly associated with spikelet and final grain number per spike, suggesting the crucial role in controlling grain yield under drought conditions. The discovered SNPs and candidate genes for drought response will be helpful for breeding drought tolerant barley cultivars.

## 5. Conclusions

Conclusively, the present study showed that drought negatively affects the yield-related traits. Despite of the reduction in most traits under drought, the heritability estimates for all respective traits were high, indicating the potential of this collection to conduct a GWAS analysis looking for drought-controlling alleles/genes. Additionally, the present study revealed that combining GWAS and bioinformatics is a very instrumental approach to identify candidate genes even for polygenic traits such as the yield-related components. Our results confirmed that the yield-related components are under polygenetic control; under contrasting growth conditions (control and drought). The candidate genes exhibited different patterns of traits control; some genes were adaptive (*HORVU2Hr1G091170*), while other genes were constitutive (*HORVU2Hr1G091030* and *HORVU3Hr1G098200*). The constitutive pleiotropic genes are of high importance to improve drought tolerance because they can be employed to improve several traits at a time with no need to test under drought. The causative genes showed different expression patterns; the constitutive genes showed constant expression profiles, while the adaptive genes showed a noisy expression profile. Most of the causative genes were expressed in spikelet organs (palea, lema and lodicules), as well as in grain, spike and leaf, indicating their potential role in drought tolerance, in particular, during the reproductive stage. To get more comprehensive and clear answers, a new detailed experiment is underway to study the gene expression of the candidate genes identified in this study especially, the gene *HORVU3Hr1G098200* because it regulates the variation of ten traits, and because of its constitutive/adaptive mode of action.

## Figures and Tables

**Figure 1 genes-11-00533-f001:**
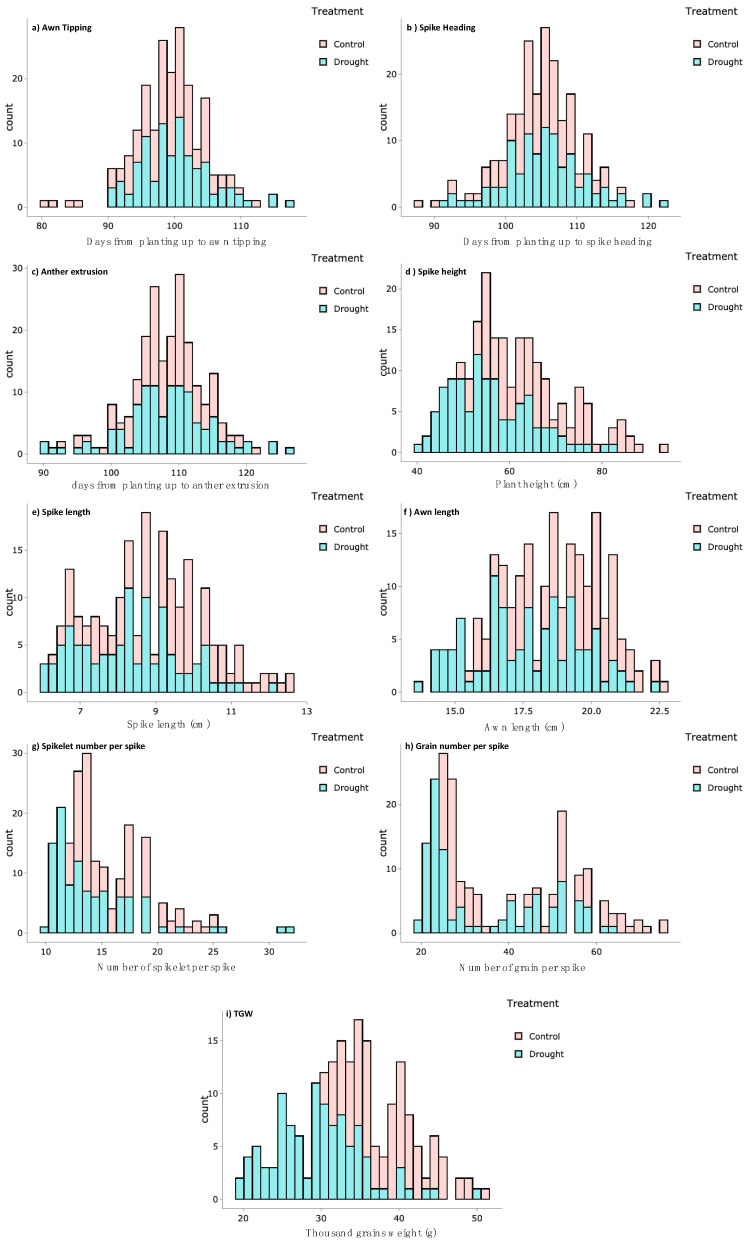
Histogram of phenotypic values distribution analysis of the studied traits; (**a**) Awn Tipping, (**b**) Spike Heading, (**c**) Anther Extrusion, (**d**) Plant Height, (**e**) Spike length, (**f**) Awn Length, (**g**) Spikelet per Spike, (**h**) Grain per Spike and (**i**) Thousand Grain Weight (TGW) in 121 spring barley accessions under control and drought stress.

**Figure 2 genes-11-00533-f002:**
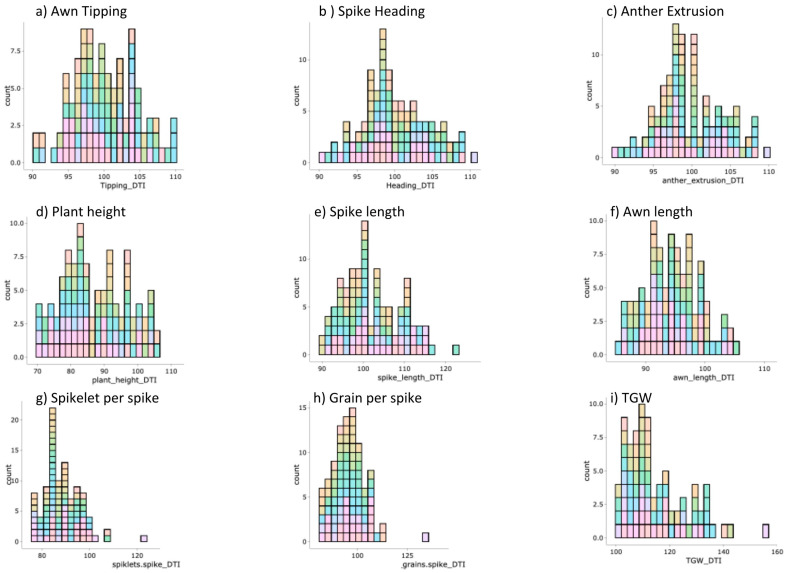
Histogram of phenotypic values distribution analysis of drought tolerance index variation of the studied traits in 121 spring barley accessions under control and drought stress. (**a**) Awn Tipping, (**b**) Spike Heading, (**c**) Anther Extrusion, (**d**) Plant Height, (**e**) Spike length, (**f**) Awn Length, (**g**) Spikelet per Spike, (**h**) Grain per Spike and (**i**) Thousand Grain Weight (TGW).

**Figure 3 genes-11-00533-f003:**
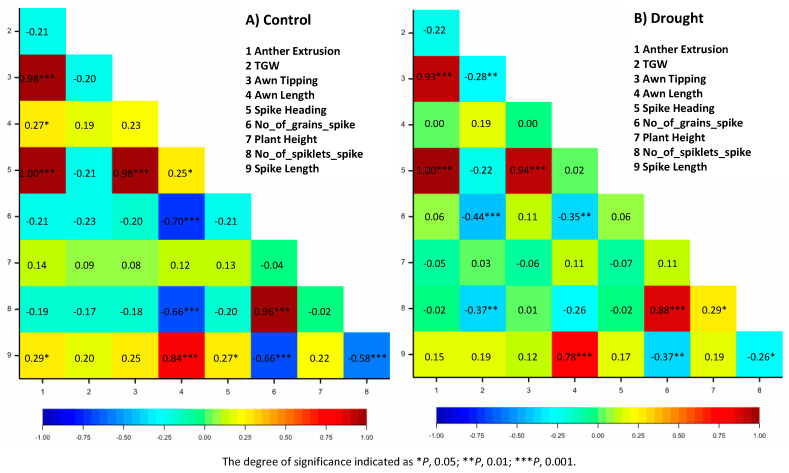
Correlations matrix among the studied traits in barley genotypes (**A**) under control, and (**B**) under drought stress. The degree of significance is indicated as * *p*, 0.05; ** *p*, 0.01; *** *p*, 0.001; ns: not significant.

**Figure 4 genes-11-00533-f004:**
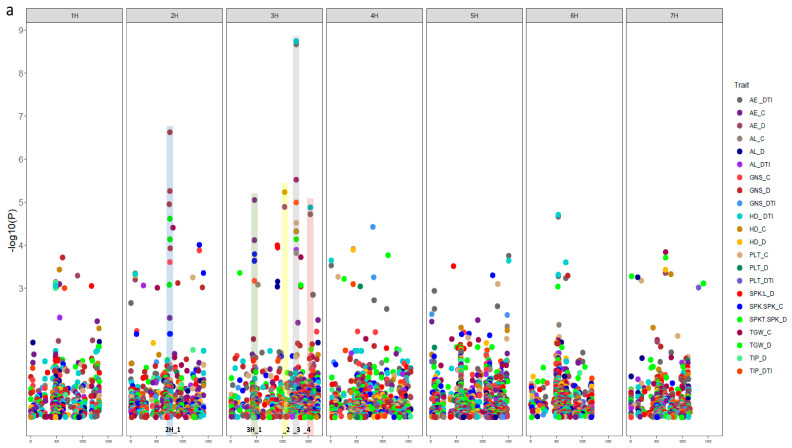
(**a**) The significant SNPs (101 SNPs, −log_10_ ≥ 3) associated with all traits under control and drought stress conditions. The x-axis shows the chromosomes and the SNP positions. The y-axis shows the −log_10_ (*P*-value) for each SNP marker. (**b**) Heatmap linkage disequilibrium to detect the region of candidate genes, which show a consistent effect on traits and associated with SNPs passing ≥ FDR using their physical position.

**Figure 5 genes-11-00533-f005:**
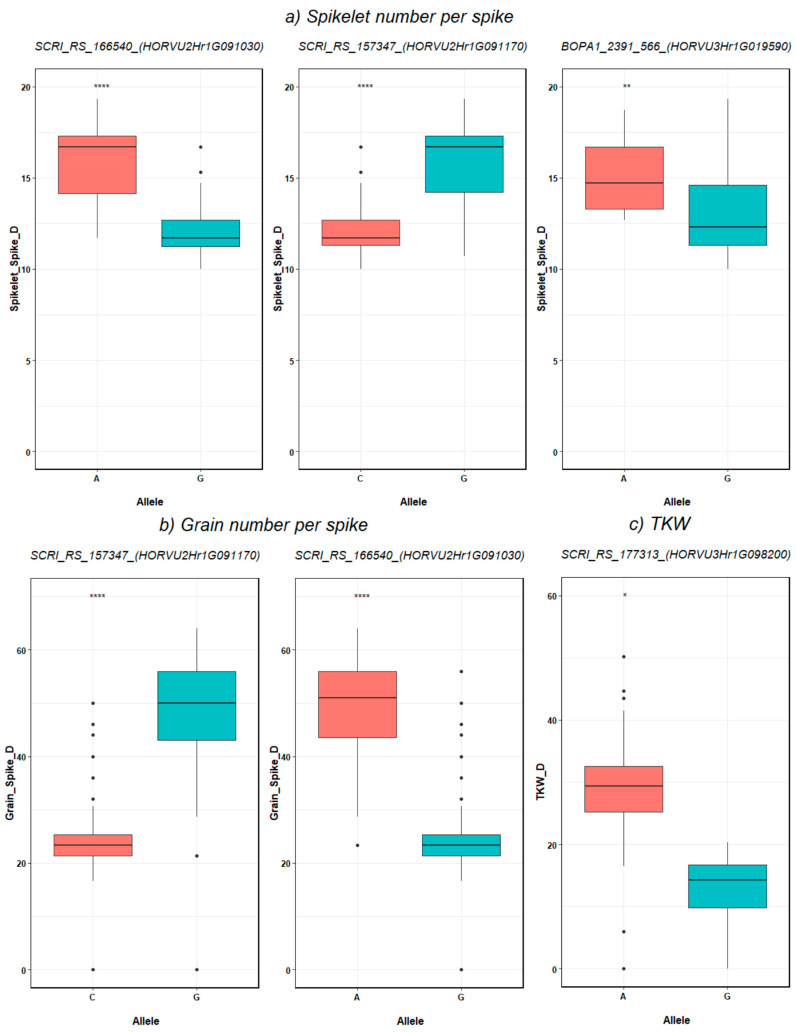
SNP-gene-based analysis for (**a**) Spikelet number per spike, (**b**) Grain number per spike in barley genotype, and (**c**) TKW (gram). The degree of significance indicated as * *p*, 0.05; ** *p*, 0.01; *** *p*, 0.001; **** *p*, 0.0001.

**Figure 6 genes-11-00533-f006:**
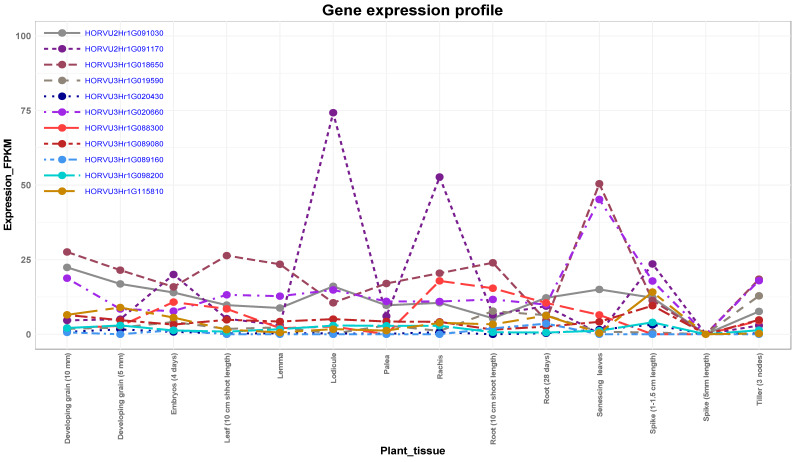
Expression analysis of the candidate genes in different plant organs at different developmental stages.

**Table 1 genes-11-00533-t001:** The name and abbreviation of measured traits and respective description of measurements.

Trait	Abbreviation		Description
	Control Drought		
Awn Tipping	AT_C	AT_D	The number of days from planting up to awn tipping.
Spike Heading	SH_C	SH_D	The number of days from planting up to spike heading
Anther Extrusion	AE_C	AE_D	The number of days from planting up to anther extrusion.
Plant Height	PH_C	PH_D	The distance between the ground level to the tip of the highest spikelet (excluding awns) in cm.
Spike Length	SL_C	SL_D	Distance from the base of the spike to the tip of the highest spikelet (excluding awns) in cm.
Awn Length	AL_C	AL_D	Distance from the tip of the spike to the end of the awn in cm.
No of Spikelets per Spike	NSS_C	NSS_D	The actual count of the number of spikelets.
No of Grains per Spike	NGS_C	NGS_D	The actual count of the number of the grains.
Thousand Grain Weight	TGW_C	TGW_D	The weight of 1000 grains randomly taken from each plot in gram (g).
Drought tolerance index (Awn Tipping)	DTI_AT		DTI $\left(\mathrm{AT}\right)=\frac{\mathrm{AT}\;\mathrm{under}\;\mathrm{drought}}{\mathrm{AT}\;\mathrm{under}\;\mathrm{control}}\times 100$
Drought tolerance index (Spike Heading)	DTI_SH		DTI $\left(\mathrm{SH}\right)=\frac{\mathrm{SH}\;\mathrm{under}\;\mathrm{drought}}{\mathrm{SH}\;\mathrm{under}\;\mathrm{control}}\times 100$
Drought tolerance index (Anther Extrusion)	DTI_AE		DTI $\left(\mathrm{AE}\right)=\frac{\mathrm{AE}\;\mathrm{under}\;\mathrm{drought}}{\mathrm{AE}\;\mathrm{under}\;\mathrm{control}}\times 100$
Drought tolerance index (Plant Height)	DTI_PH		DTI $\left(\mathrm{PH}\right)=\frac{\mathrm{PH}\;\mathrm{under}\;\mathrm{drought}}{\mathrm{PH}\;\mathrm{under}\;\mathrm{control}}\times 100$
Drought tolerance index (Spike Length)	DTI_SL		DTI $\left(\mathrm{SL}\right)=\frac{\mathrm{SL}\;\mathrm{under}\;\mathrm{drought}}{\mathrm{SL}\;\mathrm{under}\;\mathrm{control}}\times 100$
Drought tolerance index (Awn Length)	DTI_AL		DTI $\left(\mathrm{AL}\right)=\frac{\mathrm{AL}\;\mathrm{under}\;\mathrm{drought}}{\mathrm{AL}\;\mathrm{under}\;\mathrm{control}}\times 100$
Drought tolerance index (No of Spikelet per Spike)	DTI_NSS		DTI $\left(\mathrm{NSS}\right)=\frac{\mathrm{NSS}\;\mathrm{under}\;\mathrm{drought}}{\mathrm{NSS}\;\mathrm{under}\;\mathrm{control}}\times 100$
Drought tolerance index (No of Grain per Spike)	DTI_NGS		DTI $\left(\mathrm{NGS}\right)=\frac{\mathrm{NGS}\;\mathrm{under}\;\mathrm{drought}}{\mathrm{NGS}\;\mathrm{under}\;\mathrm{control}}\times 100$
Drought tolerance index (Thousand Grain Weight)	DTI_TGW		DTI $\left(\mathrm{TGW}\right)=\frac{\mathrm{TGW}\;\mathrm{under}\;\mathrm{drought}}{\mathrm{TGW}\;\mathrm{under}\;\mathrm{control}}\times 100$

**Table 2 genes-11-00533-t002:** Analysis of variance and heritability for the measured traits under control and drought treatments.

Trait	Control			*H* ^2^	Drought			*H* ^2^
	T	G	T × G		T	G	T × G	
Awn Tipping	ns	***	**	0.72	ns	**	***	0.84
Spike Heading	ns	***	**	0.80	ns	**	***	0.78
Anther Extrusion	ns	***	*	0.79	ns	*	***	0.81
Plant Height	***	***	***	0.75	***	***	***	0.71
Spike Length	***	***	***	0.74	***	***	***	0.69
Awn Length	***	***	***	0.79	***	***	***	0.76
No of Spikelet per Spike	***	***	***	0.76	***	***	***	0.65
No of Grain per Spike	***	***	***	0.77	***	***	***	0.68
Thousand Grain Weight	***	***	***	0.72	***	***	***	0.64
Awn Tipping_DTI	–	***	–	0.78				
Spike Heading_DTI	–	***	–	0.71				
Anther Extrusion_DTI	–	***	–	0.68				
Plant Height_DTI	–	***	–	0.73				
Spike Length_DTI	–	***	–	0.74				
Awn Length_DTI	–	***	–	0.71				
No of Spikelet per Spike _DTI	–	***	–	0.74				
No of Grain per Spike _DTI	–	***	–	0.68				
Thousand Grain Weight_DTI	–	***	–	0.77				

*H*^2^—Heritability; T—Treatment; G—Genotype; T × G—Treatment by Genotype interaction; DTI—Drought Tolerant Index; The degree of significance is indicated as * *p*, 0.05; ** *p*, 0.01; *** *p*, 0.001; ns: not significant.

**Table 3 genes-11-00533-t003:** The functional annotation of the putative candidate genes associated with the studied traits under drought and control growth conditions.

*Trait*	*Marker*	*Chr*	*Pos.*	*SNP Pos.*	*Gene*	*Gene Pos.*	*GO ID*	*Annotation in Barley*	*Orthologs*
NGS_CNGS_DNSS_D	SCRI_RS_166540	2H	75.56	646934425	*HORVU2Hr1G091030*	646930069-646939693	None	RNA polymerase II C-terminal domain phosphatase-like 1	AT4G21670
NGS_DNSS_D	SCRI_RS_157347	2H	75.56	647255135	*HORVU2Hr1G091170*	647252087-647258342	None	expansin B3	AT4G28250
AE_CNGS_C NSS_C	SCRI_RS_229693	3H	44.26	48643412	*HORVU3Hr1G018650*	48643412-48647662	GO:0000287GO:0003824GO:0030976	pyruvate decarboxylase-2	AT5G54960
AE_CNGS_C NSS_C	BOPA2_12_30737	3H	45.55	63626447	*HORVU3Hr1G020430*	63623393-63623651	None	Protein HASTY 1	AT3G05040
AE_CNGS_CNSS_C	BOPA1_7045-950	3H	45.55	64618663	*HORVU3Hr1G020660*	64617802-64621872	GO:0005515	Chromosome 3B, genomic scaffold, cultivar Chinese Spring	AT2G36270
AE_CNGS_CNSS_D	BOPA1_2391-566	3H	46.25	55590274	*HORVU3Hr1G019590*	55915087-55917096	GO:0003677	myb domain protein 37	AT5G23000
AE_CSH_C	SCRI_RS_230075	3H	103.27	624309908	*HORVU3Hr1G088300*	624309585-624311169	None	Chromosome 3B, genomic scaffold, cultivar Chinese Spring	AT2G36270
AE_CSH_C	BOPA2_12_30223	3H	104.32	627767267	*HORVU3Hr1G089160*	627749990-627754917	GO:0003677GO:0003700GO:0006355	AP2-like ethylene-responsive transcription factor	AT2G28550
AE_CSH_C	SCRI_RS_192360	3H	104.26	627260224	*HORVU3Hr1G089080*	627258327-627265762	None	undescribed protein	
AE_DTIAL_DTIAE_CAL_CSH_CSL_CTGW_CTGW_DSH_DTIAT_DTI	SCRI_RS_177313	3H	126.69	657713242	*HORVU3Hr1G098200*	657712024-657713975	GO:0005515	Chromosome 3B, genomic scaffold, cultivar Chinese Spring	AT2G36270
AE_DTISH_DTI	BOPA2_12_10981	3H	154.15	696452271	*HORVU3Hr1G116790*	696450874-696453390	GO:0005215GO:0006810GO:0016020	Aquaporin-like superfamily protein	AT2G36830
AE_DTISH_DTI	SCRI_RS_237738	3H	154.6	694105354	*HORVU3Hr1G115810*	694103839-694106776	None	Kinetochore protein spc25

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
