# Peer review of "Natural Variation Uncovers Candidate Genes for Barley Spikelet Number and Grain Yield under Drought Stress"

_genes, 2020, doi:10.3390/genes11050533_

Round 1
Reviewer 1 Report
There are several parts in this paper that I think requires improvement.
1) The author didn't give enough detail on how they conduct GWAS analysis, did they correct for population structure and genetic relateness among individuals, because otherwise that could lead to false positive signals.
2) The paper did not describe the genetic diversity of this association panel, for example, a PCA plot indicate the subpopulation structure should be included.
3) Although SNPs were filtered based on MAF, there might be some SNPs with high missing rate across individuals, so additional filters based on missing rate should be considered and a genome wide SNP distribution plot should be included so we can get a sense about whether there is bias in the distribution genome wide and whether some genome regions are missed due to missing data.
4) For figure 1, I would suggest presenting the result using histogram for each trait instead of boxplot. You can plot the drought and control condition side by side using the same scale so readers can get a sense that there is a large variability in this diversity panel for different traits and that's the reason doing GWAS is possible. For different index value, I would also suggest using histogram instead of boxplot, so we would be able to see that there is wide range of phenotypic data. Plotting the data using histogram also help us to visualize whether the data has a normal distribution or if there is any outlier that could be error.
5) Checking correlation between different traits is a good idea so you would expect traits that are highly correlation having overlapped genetic element involved. I also suggest adding one more correlation for the different index traits.
6) The manhattan plot in figure 4 has too much information in one plot, I would suggest having a separate plot for each trait or two, the current plot is hard to visualize.
7) For the gene expression part, I think the really important experiment that should be conducted is to check the expression of those candidate genes between drought and control conditions for drought sensitive and tolerant lines. If those genes are drought responsive, then that could serve as an additional piece of information.
8) The plot resolution is really low and there are many grammar mistakes that requires extensive editing.
Reviewer 2 Report
This manuscript attempts to identify Barley drought-stress responsive genes from 121 accessions through genome-wide association study (GWAS) strategy. From 101 significant SNP markers, eight candidate genes were located on Chromosome 2H and 3H. Additionally, author examined the tissue specific expression profile of all candidate genes. Finally, it was concluded that the identified alleles and genes would provide valuable resource for future Barley drought tolerance cultivar breeding.
Despite current study executed correctly and data was clear, there were still several issues need to be clarified.
1. Introduction, author claimed, “GWAS has been less successful for yield-related traits” (LINE 69). However, GWAS had been extensively applied to identify agronomic traits related alleles/genes in rice (Juan L Reig-Valiente et al. BMC Genomics, 2018), maize (Mona Mazaheri et al. BMC Plant Biology, 2019) and many other economic crops.
2. It would be helpful to provide pictures displaying the phenotypical changes of these Barley accessions under both control and drought stress condition.
3. Figure3, I am not sure that Thousand Grain Weight (TGW) of plants during drought stress condition is significantly (10%) higher than that of control.
4. I do not understand the expression between line 209 and 211. Is the HORVU3Hr1G098200 control specific gene, or constitutive gene? The similar confusion shown in next several paragraphs as well. Author need to rewrite these results.
5. I cannot find the RNA-seq procedure in the Material and Methods section. Author should provide comprehensive information.
6. The quality of figures is poor. Author should aim to improve them in the future. In addition, the writing and grammar have to be carefully examined as well.
Reviewer 3 Report
This manuscript describes a GWAS study in barley used to identify genes important for tolerance to drought stress during early reproduction. A pretty rigorous list of phenotypes were measured. Using 121 accessions, 101 significant SNPs were identified, with candidate genes being located on chromosomes 2 and 3. The list of genes highlighted are interesting and the authors put them into the context of what is known from model and other systems. Overall, I think this is a well-done, important study that identifies genes that are potentially key for understanding an important phenotype in an important, non-model system. Some relatively minor comments/suggestions for improvement are provided below:
- Line 83: partial sentence, so I think I am missing some methods.
- For experimental design, I know that the authors cite another publication for details, but I think a bit more information could be provided. Specifically, I am wondering how many individuals from each accession were included in this study. I am told 121 accessions and the number per pot, but not the number of plants, which makes the assessment of the statistical rigor of this study difficult. I should not have to go to another publication for this information.
- Table 2 legend: ns, DTI, T, G and H2 should be defined. No need to repeat methods, but a bit more information would make this table much more useful.
- Figure 2 is not mentioned, though I think this is just an oversite / typo. Please correct.
- Figures 4 and 5: font is VERY difficult to read (size issues).
- General question: Is there any information about the nature of the SNPs identified at the significant loci? In likely coding regions? Create knockouts or substitutions in the gene products?
- The PDF has some strange gaps. I assume this was an export problem?
- All figures and table legends: define significance (e. g. *** values). Sometimes this is included and sometimes not. Also, what do the dots in boxplots represent? (I think I know what they are, but the legend should say)
- Why are multiple sections given the designation “3.1”?
- Generally, the authors need to be careful about using undefined acronyms. The manuscript could be improved by maybe the inclusion of a list? This would greatly improve readability.
Round 2
Reviewer 1 Report
1) I suggest adding Fst calculation to indicate the genetic diversity of this population.
2) It is not clear how the authors have the genetic position instead of physical position for each SNP, is that the information from the SNP chip? please clarify. Thanks.
3) I still think the histogram distribution plot instead of bar-plot for each phenotype should go into the main body of this paper, because histogram still show you the difference between different treatments while keeping the large variation information there, which is the basis why GWAS is possible.
Author Response
Dear reviewer
thank you for your valuable comments and suggestions which are considered in the revised manuscript
Please see the attachment for the important revisions made to the manuscript based on comments as a point-by-point.
We hope these changes would be fine with you
We look forward to receiving a positive response from you
Yours sincerely
Samar Thabet and Ahmad Alqudah

Reviewer 2 Report
The authors have made significant revision according to previous comments. However, there are a few concerns remaining to be addressed. 1. Comment 3: Figure3, I am not sure that Thousand Grain Weight (TGW) of plants during drought stress condition is significantly (10%) higher than that of control. Authors’ Response: I am sorry, I could not trace this back, figure 3 deals with correlations among different traits. If this is untraceable, a clear explanation regarding the awkward difference should be involved in Discussion. In Figure 1i, the TGW under drought condition is significant less than that of control. 2. In table3, three identified genes, such as HORVU3Hr1G020660, HORVU3Hr1G088300, and HORVU3Hr1G098200, was all mapped in Chromosome 3H, and was pinpointed to the same ortholog in Arabidopsis, AT2G36270. Additional discussion should be added, regarding: a) whether they were segmental duplication during evolution; b) comparative analysis of their tissue specific expression pattern; c) based on results from a) and b), provide forward looking insights.Author Response
Dear reviewer
thank you for your valuable comments and suggestions which are considered in the revised manuscript.
Please see the attachment for the important revisions made to the manuscript based on comments as a point-by-point.
We hope these changes would be fine with you
We look forward to receiving a positive response from you
Yours sincerely
Samar Thabet and Ahmad Alqudah
